# From Circulating Tumor Cells to Mirna: New Challenges in the Diagnosis and Prognosis of Medullary Thyroid Cancer

**DOI:** 10.3390/cancers15154009

**Published:** 2023-08-07

**Authors:** Claudia Ricci, Antonia Salvemini, Cristina Dalmiglio, Maria Grazia Castagna, Silvia Cantara

**Affiliations:** Department of Medical, Surgical and Neurological Sciences, University of Siena, 53100 Siena, Italy; claudia.ricci@unisi.it (C.R.); antonia.salvemini@student.unisi.it (A.S.); cristina.dalmiglio2@unisi.it (C.D.); mariagrazia.castagna@unisi.it (M.G.C.)

**Keywords:** medullary thyroid cancer, circulating tumor cells, miRNA, cell-free DNA, extracellular vesicles

## Abstract

**Simple Summary:**

Medullary thyroid cancer derives from calcitonin-producing C cells and is characterized by sporadic and familial RET-mutated forms. Serum calcitonin represents the most reliable and sensitive marker for diagnosis and postsurgical monitoring of medullary thyroid cancer. However, in some patients, calcitonin does not achieve normal levels after surgery leading to misleading diagnosis in terms of outcome and worsening patient management. Therefore, there is a need to find new biomarkers to be used together with serum calcitonin to increase sensitivity and specificity. In this paper, we review the literature from 2010 to 2023 on circulating tumor cells, cell-free DNA, and miRNA and their application in MTC.

**Abstract:**

Medullary thyroid carcinoma (MTC) is a malignant tumor that arises from parafollicular C cells, which are responsible for producing calcitonin. The majority (75%) of MTC cases are sporadic forms, while the remaining (25%) have a hereditary component. In these hereditary cases, MTC can occur in conjunction with other endocrine disorders (i.e., pheochromocytoma) or as an isolated condition known as familial medullary thyroid carcinoma. The primary genetic mutation associated with the development of MTC, regardless of its hereditary or sporadic nature, is a point mutation in the RET gene. Evaluation of serum calcitonin levels represents the most reliable and sensitive marker for both the initial diagnosis and the postsurgical monitoring of MTC. Unfortunately, most patients do not achieve normalization of postsurgical serum calcitonin (CT) levels after surgery. Therefore, there is a need to find new biomarkers to be used with serum CT in order to increase test sensitivity and specificity. In this review, we summarize the literature from 2010 to 2023 to review the role of circulating tumor cells, cell-free DNA, and miRNA and their application in diagnosis, outcome of MTC, and response to treatments.

## 1. Introduction

Medullary thyroid cancer (MTC) is a well-differentiated thyroid tumor retaining the biochemical and pathological characteristics of the parafollicular or calcitonin-producing C cells from which it derives [1]. MTC prevalence is 5–10% of thyroid cancers with a 1–2% incidence in nodular thyroid diseases. The estimated prevalence in the general population is 1/14,300 [2]. Unlike papillary thyroid cancer and follicular carcinoma, no difference in distribution between females and males has been observed. Based on the National Comprehensive Cancer Network Guidelines, individuals diagnosed at the age of 40 or younger have a favorable prognosis, with a five-year survival rate of 95 percent and a ten-year survival rate of 75 percent. Conversely, for individuals above the age of 40, the prognosis shows a five-year survival rate of 65 percent and a ten-year survival rate of 50 percent [3]. The survival rate of MTC patients is significantly better for subjects diagnosed and treated when the tumor is limited to the thyroid. Serum calcitonin (CT) is the most specific and sensitive marker of MTC for both the primary diagnosis and the postsurgical follow-up. It is produced at significantly elevated levels in nearly 100% of primary and metastatic MTC cases. The pathogenetic mechanism has been recognized in the activation of the rearranged during transfection (RET) proto-oncogene [4,5,6,7]. Approximately 25% of cases of MTC (medullary thyroid carcinoma) are inherited as an autosomal dominant trait with varying expressivity and age-related penetrance. In these instances, there is a possibility for the involvement of other organs such as the parathyroid and adrenal glands. Additionally, this inheritance pattern gives rise to three distinct syndromes known as multiple endocrine neoplasia (MEN) 2A, MEN 2B, and familial medullary thyroid carcinoma (FMTC) [8,9,10]. A better outcome is observed in hereditary MTC cases detected using genetic screening of the RET proto-oncogene and treated in a preclinical phase when the tumor is microscopic and confined to the thyroid [11]. The other 75% of the cases are represented by sporadic forms characterized by somatic point mutations in the RET gene [12]. The RET proto-oncogene is located on chromosome 10q11-2 and encodes a receptor tyrosine kinase that binds to extracellular signaling molecules from the glial cell line-derived neurotrophic factor family [13,14]. Several mutations are known to be associated with MEN 2A. These mutations affect the cysteine-rich extracellular domain, in particular, codons 609 (p.C609X), 611 (p.C611Y), 618 (p.C618X), 620 (p.C620X), and 630 (p.C630R), and in 95% of the cases, the codon 634 (p.C634R) [15]. Another mutation causes more than 90% of MEN 2B; this change replaces the amino acid methionine with the amino acid threonine at position 918 (p.M918T) [16]. Several amino acid substitutions can cause familial medullary thyroid carcinoma such as mutations at codons 609 (p.C609X), 611 (p.C611Y), 618 (p.C618X), 620 (p.C620X), 630 (p.C630R), 634 (p.C634R), 768 (p.E768D), 790 (p.L790F), 804 (p.V804L/M), 806 (p.Y806C), 883 (p.A883F), 891 (p.S891A), and 918 (p.M918T) [17]. Sporadic forms are reported to be mutated at the somatic level, especially at codons 632 (p.E632X), 633 (p.L633X), and 918 (p.M918T) (Figure 1). Mutations in HRAS and KRAS have been identified in sporadic forms as well [18]. Total thyroidectomy with prophylactic/therapeutic central compartment lymph node dissection is the initial treatment of choice [12]. The appropriate course of treatment depends on the tumor burden and rate of progression. In cases where there are only a few local or distant metastases and the rate of progression is slow, surgical treatments and local therapies are recommended. On the other hand, when there is a significant number of metastases and the disease is progressing rapidly, systemic treatments should be initiated [12].

As mentioned above, serum CT is a reliable and sensitive marker for both the initial diagnosis and the postsurgical monitoring of MTC. Unfortunately, most patients with MTC do not achieve normalization of postsurgical serum calcitonin (CT) levels or a definitive cure [19]. Therefore, there is a need to find new biomarkers to be used with serum CT in order to increase test sensitivity and specificity. In this review, we focus on approximately the last 10 years (2010–2023) of research on circulating tumor cells (CTCs), cell-free DNA (cfDNA), and miRNA and their application in MTC with the aim to identify potential limitations and strengths of these technologies in diagnosis, prognosis, and response to therapy.

## 2. Circulating Tumor Cells

Circulating tumor cells (CTCs) are rare and heterogeneous (epithelial, mesenchymal CTC (MCTC), and both mixed types) cells that are shed from primary or metastatic tumor lesions into the bloodstream [20]. They are an important biomarker for cancer, as their presence in the bloodstream indicates that cancer has spread beyond its original site and has the potential to metastasize to other parts of the body [20] (Figure 2).

The detection of CTCs is a non-invasive method and can be used for early diagnosis of cancer. It also provides important data about the biology of cancer (i.e., origin), its metastatic potential (correlated with the grade of epithelial-to-mesenchymal transition and stemness), response to treatment, and molecular characteristics [21]. This information greatly contributes to the evaluation of the treatment of choice for a specific patient in terms of response and precision medicine. In some studies [22], CTCs have been associated with tumor stage, tumor progression, and mortality. CTCs can be isolated from the bloodstream using a variety of techniques, including immunomagnetic separation, density gradient centrifugation, and microfluidic devices [20]. Once isolated, CTCs can be characterized using immunofluorescence staining, RNA sequencing, or whole-genome sequencing. These methods can provide information about the expression of specific biomarkers, mutations, and gene expression profiles in CTCs [20].

CTC evaluation has been investigated also in MTC with promising results (Table 1). In 2022, in a cohort of 12 MTC patients out of 164 patients (7.3%), Weng and colleagues [23] evaluated the number of CTCs at diagnosis using the CanPatrol capture technique and found that 58.3% (7/12 MTC patients) were CTC-positive while 5/12 (41.6%) were not. The authors considered a cutoff of six CTCs and found that, in that case, patients had shorter progression-free survival and an increased metastasis rate. The paper by Weng and colleagues strongly validates the use of CTCs as a prognosis biomarker once a clear cutoff is confirmed. In the same year, a study involving 30 MTC patients out of 394 total subjects (7.6%) validated a six-cell cutoff in an independent series. The authors showed that in MTC, overall survival (OS) was found to be significantly shorter (*p* < 0.01) in individuals with more than six CTCs compared to those with six or fewer CTCs [24].

This conclusion was drawn based on an analysis of Kaplan–Meier survival curves. The hazard ratio (HR) was determined to be 3.762, indicating a higher risk of mortality, with a 95% confidence interval (CI) ranging from 1.299 to 10.89 [24]. Similar results were reported in another study [25] by Xu et al. They analyzed 18 MTC patients with distant metastasis and found that 6/18 (33.4%) had five or more CTCs, which were detected using the CellSearch assay. The median survival time for patients with five or more CTCs was 13 months, compared to 51.5 months for those with fewer than five CTCs (*p* = 0.0116). This significant difference in median survival indicates that a higher CTC count is associated with a poorer prognosis in metastatic MTC patients. Moreover, the use of a different technique allowed for lowering the cutoff from six to five CTCs. We mentioned that calcitonin (CT) is a sensitive and accurate marker for detecting the recurrence of medullary thyroid carcinoma following thyroidectomy. However, it is important to note that there are potential limitations associated with calcitonin immunoassays due to the presence of heterophilic antibodies or macroaggregates that can lead to falsely increased values [28]. Additionally, in some cases, calcitonin levels may remain undetectable despite the presence of metastasis. In this view, the presence of calcitonin-positive CTCs might contribute to patient management. One study from 2018 [26] searched for these types of CTCs using an optimized technique in nine MTCs with a serum calcitonin concentration ranging from 48 to 10,600 pg/mL. Calcitonin-positive CTCs were identified in eight out of nine (89%) patients, but the CTC counts did not always align with the TNM staging system. The authors found that one patient classified as pT3N1bM1 did not exhibit detectable calcitonin-positive CTCs, while another subject, classified as pT1N1bMx, had five CTCs [26]. Even a correlation between serum calcitonin concentrations and CTC counts was missing [26]. On the contrary, a correlation between the number of CTCs and the T stage at initial diagnosis (Pearson r = 0.9837, *p* = 0.0163) was found in a study by Ehlers and colleagues [27]. However, neither lymph node nor distant metastasis nor CT levels had an impact on the number of CTCs in that study [27].

The emerging message from these studies is that the search for CTCs in MTC is feasible and represents a specific test. An elevated number (from four to >six) of circulating tumor cells in patients with MTC may serve as a robust biomarker for predicting cancer prognosis independently from the method used for their extraction and characterization (Table 1) and may improve patient outcomes using more personalized and targeted treatments. Nevertheless, more efforts are needed to find a clear cutoff that is able to detect disease recurrence even many years after surgery and particularly in CT-negative cases.

## 3. Cell-Free DNA

Cell-free DNA (cfDNA) derived from tumors is a promising biomarker for cancer diagnosis and monitoring. It consists of small fragments of DNA released into the bloodstream by tumor cells. Analyzing cfDNA can provide valuable information about tumor genetic alterations, such as mutations, copy number variations, and epigenetic modifications. cfDNA testing is minimally invasive as it can be obtained with a simple blood draw. The detection and analysis of tumor-derived cfDNA can help in early cancer detection, treatment selection, and monitoring of the treatment response. However, cfDNA testing faces challenges, including the low abundance of tumor-derived cfDNA, the presence of normal DNA, and technical limitations in detecting rare mutations. Ongoing research aims to improve the sensitivity and specificity of cfDNA-based tests to maximize their clinical utility in various cancer types including MTC. Most of the studies concentrate on the mutational profile of the cfDNA. By analyzing 29 MTC patients, Ciampi and colleagues revealed that 4/26 (15.4%) cases showed positive pre-operative cfDNA with a significantly higher presence of RET p.M918T mutation (*p* = 0.0468) and a higher frequency of persistent disease [28]. Three patients had positive post-operative cfDNA (but not pre-operative), and seven cases with persistent disease harbored either pre- or post-operative positive cfDNA [29]. In all cases, free DNA correlated with serum CT [28], but pre-operative cfDNA was not useful for diagnostic purposes, although it was for outcome prediction, while post-operative cfDNA showed significant results in response to therapy monitoring. In a series composed of both MTC and differentiated thyroid cancer, Allin et al. [30] found that most patients (67%) with advanced thyroid cancer had earlier detectable cfDNA. The detection rate was higher in MTC compared with PTC and FTC, and the detection rate was higher in patients with metastatic disease (79%) compared with those with local recurrence (33%) and no macroscopic disease (0%), indicating that cfDNA correlates with disease aggressiveness. In their series, early disease progression was found in 3/15 (20%) MTC cases. Again, cfDNA correlated with serum CT [30].

When compared to healthy control subjects (n = 19), cfDNA showed a high ability to diagnose MTC (n = 58) [31]. In that paper, positive values of Ct corresponded to a lower amount of cf-DNA, and the same was found for RET mutational status. These findings might suggest that cfDNA represents a potential substitute marker for MTC in the case that the classic markers, such as CT and RET, are negative [31]. But might cfDNA predict the overall survival of MTC patients with better reliability than calcitonin? Cote et al. [32] tried to answer this question. They analyzed 75 patients with confirmed sporadic MTC diagnosis, a serum calcitonin measurement > 100 pg/mL, and a tumor tissue harboring the RET p.M918T mutation (50/75). In 16/50 (32%) patients, cfDNA showed the same mutation, which effectively correlated with worse overall survival [32]. In addition, the detection of RET p.M918T cfDNA was strongly associated with a worse outcome than CT doubling time [32]. The diagnostic role of cfDNA was also evaluated in 21 MTC cases in a study by Higazi [33], in which the authors measured the cfDNA integrity using specific markers (Alu244/Alu83 ratio) and found that cfDNA integrity was more elevated in MTC compared to other thyroid malignancies [33]. Taken together, these studies (Figure 3) seem to indicate that cell-free DNA from tumor origin can be used either for diagnostic or prognostic purposes, but its efficacy is increased when used for predicting disease outcome and response to therapy, especially in cases that are negative for CT and/or RET mutational status. Again, a major issue that researchers have to address is the existence (or not) of a cutoff able to predict disease recurrence with high accuracy. Cell-free DNA seems to better correlate with serum CT, so the use of cfDNA in combination with CTCs might be useful for CT-negative cases.

## 4. MicroRNAs (miRNAs)

In the last years, miRNAs have raised great interest as potential biomarkers, particularly in cancer research.

miRNAs are short non-coding RNA molecules (about 22 nucleotides in length) that act as endogenous regulators of gene expression at the post-transcriptional level. Mature miRNAs derive from primary miRNAs, which are transcribed in the nucleus and subsequently processed into pre-miRNAs by Drosha. They are then exported to the cytoplasm, where are finally cleaved by the Dicer complex, resulting in mature miRNAs [34].

miRNAs have a tissue-specific expression and play important roles in many physiological and pathological processes, including tumorigenesis [35,36]. miRNAs that are released from the cells in the bloodstream are referred to as circulating miRNAs. Circulating miRNAs can be achieved with minimally invasive approaches and have several intrinsic features, making them interesting as biomarkers. miRNAs have been shown to have sensitivity, specificity, and predictive power [37,38]. In addition, in contrast to other RNA classes, they are very stable and can be reliably quantified in the blood [39]. Moreover, miRNAs detected in biological fluids can “mirror” changes in the cells of origin, and their levels may be associated with specific clinical features, responses to treatments, and patient outcomes [40].

### 4.1. miRNA Detection

Despite the characteristics that make miRNAs suitable biomarkers, detecting these molecules may be challenging due to their intrinsic characteristics, such as small size, low levels of expression, sequence similarity among different members, and tissue- or developmental stage-specific expression. Two general approaches are commonly used in the research of miRNAs:(1)The use of profiling methods, such as microarray, quantitative real-time polymerase chain reaction (qRT-PCR)-based array, quantitative nCounter, or next-generation sequencing (NGS), to quantify hundreds of miRNAs in samples obtained from patients with a pathology of interest in comparison to control subjects. This approach is usually followed by the validation of identified miRNAs using qRT-PCR or other techniques.(2)The selection of a subset of specific miRNAs related to specific tissues, cells, gene expression pathways, and diseases. In this case, the number of miRNAs to study is limited, and other experimental approaches can be used. These include individual qRT-PCR, which allows increasing sensitivity and reproducibility of the analysis, and droplet digital PCR technology, which provides miRNA absolute quantification without the requirement of standard curves, efficiency correction methods, or technical replicates [41].

Considering the wide range of possible approaches to miRNA research, it is not surprising that different studies often obtain conflicting results, and a comparison among them is quite difficult. In addition, circulating miRNAs are present in the body fluids freely circulating, bound to specific proteins, associated with lipoproteins, and enclosed in extracellular vesicles (EVs) [42]. EVs, which include exosomes and microvesicles, are heterogeneous groups of vesicles that are released by almost all cell types. They consist of a lipid membrane, contain specific molecules, and constitute a communication system between different cell types in physiological and pathological conditions [43]. Circulating pEV miRNAs may represent an interesting source of information since they play a fundamental role in cell–cell communication, are associated with specific pathological conditions, are easy to detect in blood samples, and are not affected by hemolysis, which can alter miRNA expression levels. Selecting the fractions of circulating miRNAs to study (free circulating versus EV miRNAs or total plasma miRNAs) is another source of heterogeneity in this kind of research.

### 4.2. Circulating miRNAs in MTC

To date, six studies have been published exploring miRNA expression levels in plasma and serum samples obtained from MTC patients. The results are summarized in Table 2.

Romeo et al. identified increased levels of plasma miR-375 in MTC in comparison to healthy controls and subjects in remission. Circulating miR-375 levels were higher in male compared to female patients, while this difference was not present in matched healthy controls. Importantly, high levels were associated with distant metastases and reduced overall survival and were a strong prognostic marker of poor prognosis in MTC patients. In addition, although miR-375 plasma levels did not reach statistical significance as a predictor of the vandetanib response, the trend observed suggested that miR-375 could be considered as a possible biomarker for the response to treatments in future studies [44].

Zhang et al. showed that the serum levels of miR-222-3p and miR-17-5p were significantly augmented in MTC patients compared to patients with benign nodules and healthy control subjects. Receiver operating characteristic (ROC) curve analyses confirmed the high diagnostic accuracy of the two miRNAs, suggesting they can represent potential diagnostic biomarkers in MTC, especially when used in combination [45].

Shabani et al. examined plasma levels of miR-144 and miR-34a, in MTC patients with or without RET mutations and in healthy controls. Both miRNAs were expressed at higher levels in patients with MTC than in controls and in patients carrying RET mutation than in RET wild-type patients. However, a ROC curve analysis showed that specificity and sensitivity are not suitable for using miR-144 and miR-34a as circulating MTC biomarkers [46].

Censi et al. further assessed the diagnostic and prognostic value of circulating miR-375 levels in MTC, showing that the levels of this miRNA were 101 times higher in the serum of MTC patients than in all the other patients and controls included in the study, without overlap. However, no correlation between serum and tissue miR-375 levels was observed. In addition, serum miR-375 and CT levels showed a negative correlation in MTC patients. The ROC curve analysis showed that serum miR-375 levels can be used as a biomarker in MTC diagnosis, with a notable specificity [47].

Melone et al. applied miRNome profiling in the context of a wider multiomics approach, aimed at identifying common molecular and functional signatures of different neuroendocrine neoplasms (NENs), including MTCs. They identified a set of 13 miRNAs that could be evaluated as possible circulating biomarkers. Overall, these miRNAs were significantly overexpressed in NEN patients compared to healthy subjects. Among them, miR144-3p, miR7-5p, and miR335-5p were significantly upregulated in MTC. Furthermore, miR-375 was significantly upregulated in all the different NEN subgroups [48].

Finally, Besharat et al. focused on miRNAs obtained from plasma extracellular vesicle samples. Using miRNA arrays in the first cohort of MTC patients, they identified a set of circulating miRNAs as diagnostic biomarkers. Among them, miR-26b-5p and miR-451a showed a higher expression, which decreased during follow-up in disease-free patients. The diagnostic performance of the two miRNAs was validated in a second independent cohort of MTC patients and control subjects using digital PCR. ROC curve analyses confirmed the high diagnostic role of miR-26b-5p and miR-451a [49].

Despite a low overlapping among the miRNAs identified, some circulating miRNAs seem to be particularly promising as potential biomarkers in MTC patients. miR-375 is one of the most interesting miRNAs since its upregulation has been reported in three different studies [44,47,48] both in plasma and serum of MTC patients. It has shown high diagnostic and prognostic accuracy. In addition, miR-375 was significantly upregulated in different subgroups of NENs, suggesting that its overexpression may be a common trait of neuroendocrine cancers. miR-144 levels were also reported as being increased both in plasma and serum [46,48]. However, in this case, specificity and sensitivity were too low to propose this miRNA as a new circulating biomarker for MTC. Worth noting, only one study has been performed to date on plasma extracellular vesicles (pEVs) [49], which reported an upregulation in miR-26b-5p and miR-451a in MTC patients with significant diagnostic performance. These miRNAs had not been previously identified, probably due to the different biological samples used in other studies. In addition, the use of pEVs can avoid the effect of hemolysis, which is known to affect the expression levels of these miRNAs [50].

## 5. Conclusions and Future Perspectives

Early detection and correct treatment of MTC can allow better management of the disease. Identification of effective circulating biomarkers can provide a valuable diagnostic, prognostic, and monitoring tool. Considering the low prevalence of this kind of cancer, this research is rather problematic, and only a few biomarkers have shown to be strong and reproducible.

The literature reviewed in this article suggests that all CTCs, cfDNA, and miRNAs may represent useful biomarkers to be used for MTC.

Some of these biomarkers may be used in the diagnosis of MTC. In particular, cell-free DNA from tumor origin and some miRNAs (miR-375, miR-144 and miR-34a, miR-26b-5p, and miR-451a) are able to effectively distinguish MTC patients from control subjects.

Other molecules have shown a robust role as prognostic markers. In this case, CTCs, cell-free DNA from tumor origin, miR-375, miR-26b-5p, and miR-451a can represent valuable tools to predict disease outcome and/or response to therapy.

Each of these categories of biomarkers has particular strengths and weaknesses that must be considered.

Particularly, the search for CTCs in patients with MTC has proven to be feasible and represents a specific test. A number of circulating tumor cells from four to more than six may provide a robust biomarker for predicting cancer prognosis and improving patient outcomes using personalized and targeted treatments. A point of strength of this test is that it is independent of the method used for extraction and characterization. On the other hand, a clear cutoff for the cell number that is able to detect disease recurrence even many years after surgery and particularly in CT-negative cases has still to be univocally identified.

Cell-free DNA from tumor origin can be used either for diagnostic or prognostic purposes, in particular, for predicting disease outcome and response to therapy in cases negative for CT and/or RET mutations, where their efficacy is increased. Also, in this case, a major issue is the identification of a cutoff able to predict disease recurrence with high accuracy. Interestingly, cell-free DNA seems to correlate well with serum CT, so its use in combination with CTCs might be useful for CT-negative cases.

Finally, miRNAs, especially miRNA-375, showed high accuracy both in MTC diagnosis and prognosis. This miRNA has been identified both in the plasma and serum of MTC patients. It has the potential to be particularly useful when CT levels are moderately high and a thyroid nodular disease is suspected. Furthermore, miR-375 analysis seems to also be a promising prognostic biomarker in patients undergoing tyrosine kinase inhibitor (TKI) treatments, although validation of these results using a larger cohort is necessary.

Some critical points are common to the study CTCs, cfDNA, and miRNAs as biomarkers for MTC, and specific improvements in the research approach should be taken into account to overcome them. First, the comparison among data reported by different research groups is often complicated by the wide range of methods used to identify and measure potential biomarkers. A general formulation of accepted guidelines, standard protocols, and strong methods for statistical analysis could allow for achieving reliable biomarkers. For example, in the case of miRNA studies, a critical issue is proper normalization for accurate miRNA quantification. Several methods, such as using external miRNAs, evaluating endogenous miRNAs or small RNAs, and normalizing miRNA concentration to serum volume, are generally used. Since normalization is essential to avoid inaccurate quantification, stable normalizers are needed, but their identification is often challenging.

Another critical point is the relatively small number of patients included in some studies carried out until now. Even though the results are often promising, they need to be verified using larger cohorts of MTC patients, especially when the correlation with disease outcome and response to therapies are considered. Thus, those biomarkers, which have shown to be promising in early studies, should be validated in independent laboratories and/or in multicenter collaborations.

Finally, one interesting approach to biomarker study is the evaluation of a complex set of markers, or combinations or ratios of molecules of different origins, rather than the use of a single biomarker. This strategy has been shown to improve the sensitivity and/or specificity of potential biomarkers and provide more exhaustive diagnostic/prognostic information, and thus it should be used in future research on MTC biomarkers.

In conclusion, CTCs, cfDNA, and miRNAs represent promising biomarkers for MTC, and their use, alone or in combination, may improve patient outcomes using more personalized and targeted treatments. The ultimate objective is to include these biomarkers in all phases of patients’ management, from diagnosis to follow-up and, in perspective, also in clinical trials to identify future therapeutic approaches.

## Figures and Tables

**Figure 1 cancers-15-04009-f001:**
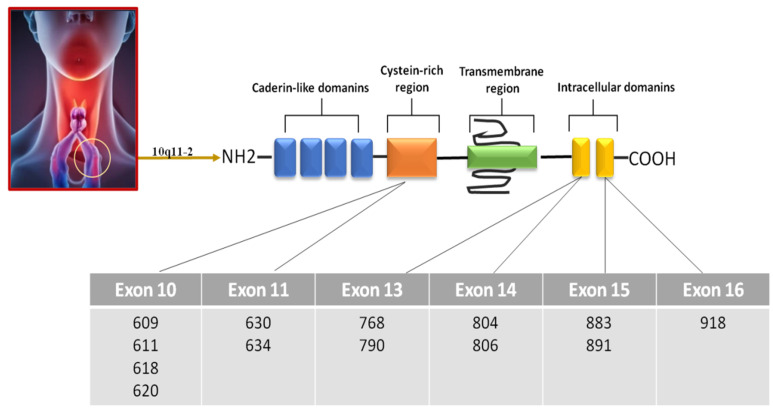
RET gene domains and different mutation sites.

**Figure 2 cancers-15-04009-f002:**
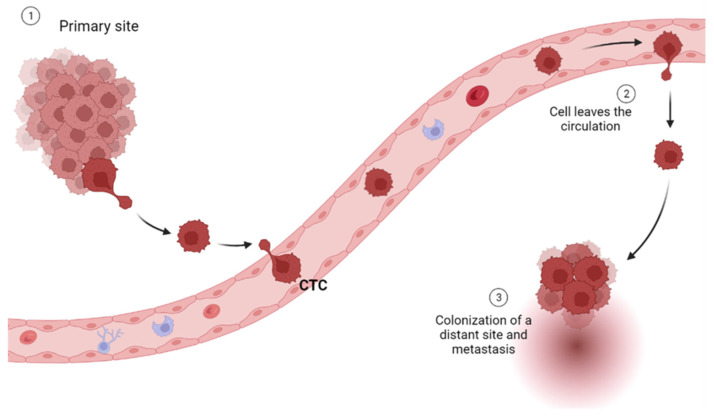
Graphical representation of distant site colonization by a circulating tumor cell (CTC) detached from a primary site (created in BioRender.com).

**Figure 3 cancers-15-04009-f003:**
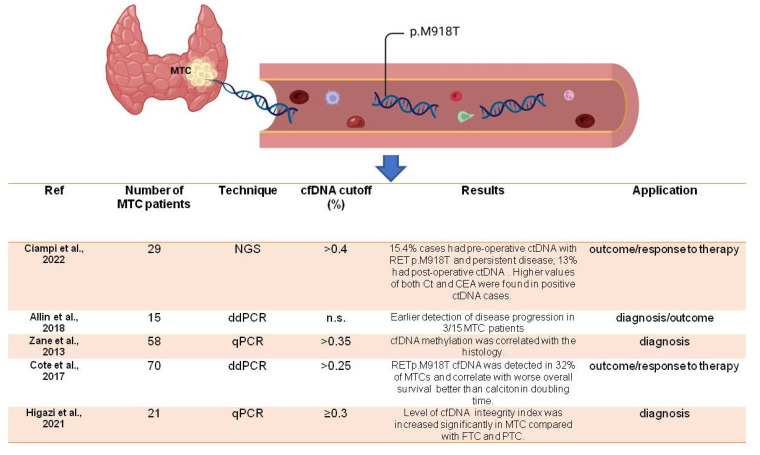
p.M918T represents the most common mutation found in cfDNA from medullary thyroid cancer. The scheme reports study results and the role of cfDNA in diagnosis or in outcome [29,30,31,32,33].

**Table 1 cancers-15-04009-t001:** Summary of the analyzed studies on CTCs.

Ref.	Number of Analyzed MTC Patients	Technique	Peripheral Blood mL	Results
[23]	12	CanPatrol capture	5	7/12 (58.3%) CTC-positive 5/12 (41.6%) CTC-negative
[24]	30	CanPatrol capture	5	>6 CTCs worst prognosis (up to 60 months)
[25]	18	Veridex Cell Search	7.5	>5 CTCs worst prognosis
[26]	9	ScreenCell device	7.5	8/9 (89%) CTC-positive
[27]	14	BD Vacutainer Cell Preparation Tubes	8	±5 cells in MTC patients

**Table 2 cancers-15-04009-t002:** Deregulated circulating microRNAs (miRNAs) in MTC patients.

miRNA	Source	Sample Size	Approach	Results	Ref.
miR-375	Plasma	Discovery cohort: 19 MTC patients 19 healthy control subjects Validation cohort: 17 MTC patients 17 healthy control subjects Vandetanib treatment: 20 MTC patients	Microarray profiling qRT-PCR validation	miR-375 was up-regulated in MTC patients vs. healthy controls ( AUC 0.85 for the discovery cohort and 0.75 for the validation cohort); High levels were associated with a significantly reduced overall survival (HR = 10.61, *p* < 0.0001); miR-375 plasma levels were not predictive of vandetanib response.	[44]
miR-222-3p miR-17-5p	Serum	15 MTC patients 91 patients with benign nodules 89 healthy controls	TaqMan low-density array qRT-PCR validation	miR-222-3p and miR-17-5p were significantly increased in the MTC group vs. benign nodule (AUC = 0.907) and control group (AUC = 1.000); They may serve as auxiliary tools for diagnosing MTC.	[45]
miR-144 miR-34	Plasma	50 MTC patients (25 RET-positive and 25 RET-negative) 50 control subjects	qRT-PCR	miR-144 and miR-34a were up-regulated in MTC patients vs. healthy controls, with higher levels in RET-positive patients; However, they showed no significant prognostic value as MTC biomarkers.	[46]
miR-375	Serum	69 MTC patients 49 patients with non-C cell 14 patients with pheochromocytoma 19 healthy controls	qRT-PCR	miR-375 levels were >100 times higher in MTC patients than in all other patients and controls; Negative correlation between miR-375 and CT miR-375 levels may be used as a marker in MTC diagnosis, with a high specificity (AUC = 0.978).	[47]
miR-144-3p miR-7-5p miR-335-5p miR-375	Serum	2 MTC patients 34 healthy controls	Small RNA-seq qRT-PCR	miR144-3p, miR7-5p and miR335-5p were upregulated in MTC; miR-375 levels were significantly increased in all the different NEN subgroups.	[48]
miR-26b-5p miR-451a	pEV	Discovery cohort: 25 MTC patients 22 healthy controls Validation cohort: 12 MTC patients 13 control subjects	MiRNA-TaqMan array Digital PCR	miR-26b-5p and miR-451 were highly expressed in MTC patients; Their expression decreased during follow-up in disease-free patients; ROC curve analyses showed a high diagnostic and post-surgery role (AUC = 0.87 for miR-26b-5p and 0.83 for miR-451).	[49]

Abbreviations: Ref., reference; pEV, plasma extracellular vesicle; MTC, medullary thyroid cancer; qRT-PCR, quantitative real-time polymerase chain reaction.

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
