# Peer review of "From Circulating Tumor Cells to Mirna: New Challenges in the Diagnosis and Prognosis of Medullary Thyroid Cancer"

_cancers, 2023, doi:10.3390/cancers15154009_

Round 1

Reviewer 1 Report

Ricci and colleagues summarized the role of microRNAs as a potential biomarker both in the diagnosis and prognosis of medullary thyroid cancer .

The present review article is well written; however there are some issues that must be improved .

The final aim of this review should be clearly written and not hidden; please updated this point.

The introduction is too long particularly in the description of the RET proto-oncogene . Please improved this section .

The description of circulating tumor cells is not very relevant and the clinical impact is not well explained .

The last and the most important section is too poor in the meaning and the translational correlation with the use of pre-clinical studies .

The authors should made a huge effort to try not only to describe and summarized the current litterature, but also to make their original point of view .

Author Response

Ricci and colleagues summarized the role of microRNAs as a potential biomarker both in the diagnosis and prognosis of medullary thyroid cancer .

The present review article is well written; however there are some issues that must be improved .

1) The final aim of this review should be clearly written and not hidden; please updated this point.

The last part of introduction has been modified to clearly state the aim of the review.

2) The introduction is too long particularly in the description of the RET proto-oncogene . Please improved this section.

We have shortened the introduction, eliminating not necessary details on RET proto-oncogene.

3) The description of circulating tumor cells is not very relevant and the clinical impact is not well explained .

Some information on CTCs clinical impact have been added: The detection of CTCs is a noninvasive method and can be used for early diagnosis of cancer. It also provides important data about the biology of cancer (i.e. origin), its metastatic potential (correlated with grade of epithelial-to-mesenchymal transition and stemness), response to treatment, and molecular characteristics [21]. This information greatly contributes to the evaluation of the treatment of choice for a specific patient in terms of response and precision medicine. In some studies, CTCs have been associated with tumor stage, tumor progression and mortality.

A new reference has been added: " Lin D, Shen L, Luo M, Zhang K, Li J, Yang Q, Zhu F, Zhou D, Zheng S, Chen Y, Zhou J. Circulating tumor cells: biology and clinical significance. Signal Transduct Target Ther. 2021 Nov 22;6(1):404. doi: 10.1038/s41392-021-00817-8. PMID: 34803167; PMCID: PMC8606574".

4) The last and the most important section is too poor in the meaning and the translational correlation with the use of pre-clinical studies .

As suggested by the reviewer, the last paragraph has been improved to better show the future perspectives of this kind of studies.

5) The authors should made a huge effort to try not only to describe and summarized the current literature, but also to make their original point of view .

According with referee suggestion, each section has been implemented trying to offer to the reader not only an analysis of the literature but also a critical point of view on perspectives and limitations of the methods described.

Reviewer 2 Report

This study is interesting with clinical significance.  Circulating tumor cells, cell free DNA and miRNA are important factors in in diagnosis of cancer. The authors made a comprehensive summary in the diagnosis and Prognosis of Medullary Thyroid Cancer. This is an excellent review with a strong focus and comprehensive scientific evidences. The followings are some comments to the authors.

Comments

1. I suggest adding results in the table of Figure 3 .

2. I suggest adding sample size and detailed result information of each study in the table 2 .For example, diagnostic accuracy, decreased overall survival.

3. Whether these three methods are limited to the pathological grade and clinical stage? Which of these three methods is the most promising for clinical application?

4. The discussion and conclusion can be improved. These kinds of studies have limitations. Hence, the author should have stated the potential limitations and suggested what could be done the next step in this area of research.

Author Response

This study is interesting with clinical significance.  Circulating tumor cells, cell free DNA and miRNA are important factors in in diagnosis of cancer. The authors made a comprehensive summary in the diagnosis and Prognosis of Medullary Thyroid Cancer. This is an excellent review with a strong focus and comprehensive scientific evidences. The followings are some comments to the authors.

Comments

  1. I suggest adding results in the table of Figure 3.

As suggested by the referee, results from each study have been added in Figure 3.

  1. I suggest adding sample size and detailed result informationof each study in the table 2 . For example, diagnostic accuracy, decreased overall survival.

The requested information has been added in Table 2.

3-4. Whether these three methods are limited to the pathological grade and clinical stage? Which of these three methods is the most promising for clinical application?

The discussion and conclusion can be improved. These kinds of studies have limitations. Hence, the author should have stated the potential limitations and suggested what could be done the next step in this area of research.

We thank the reviewer for these suggestions. A critical and more accurate discussion of potential applications, limitations and strengths of each biomarker has been included in the last paragraph of the revised version of the manuscript.

Round 2

Reviewer 1 Report

Thanks a lot to the authors for the new version of the manuscript.

The paper is now significantly improved.

All the points are better clarified and substantially changed the meaning of the review.

There are no critical points.